



# Forcing mechanisms of the quarterdiurnal tide

Christoph Geißler [1], Christoph Jacobi [1], and Friederike Lilienthal [1]

[1]Institute for Meteorology, Universität Leipzig, Stephanstr. 3, 04103 Leipzig, Germany

**Correspondence:** Christoph Geißler (christoph.geissler@uni-leipzig.de)

**Abstract.**

We used a nonlinear mechanistic global circulation model to analyze the migrating quarterdiurnal tide (QDT) in the middle atmosphere with focus on its possible forcing mechanisms. These are absorption of solar radiation by ozone and water vapor, nonlinear tidal interactions, and gravity wave-tide interactions. We show a climatology of the QDT amplitudes, and we exam-

ined the contribution of the different forcing mechanisms on the QDT amplitude. To this end, we first extracted the QDT in the model tendency terms. Then, we separately removed the QDT contribution in different tendency terms. We find that the solar forcing mechanism is the most important one for the QDT, but also the nonlinear and gravity wave forcing mechanism play a role in certain seasons, latitudes and altitudes. Furthermore, destructive interference between the individual forcing mechanisms are observed. Therefore, tidal amplitudes partly become even larger in simulations with removed nonlinear or gravity

wave forcing mechanism.

## 1 Introduction

The dynamics of the upper mesosphere and lower thermosphere (MLT) are strongly influenced through atmospheric waves, especially solar tides (Yiğit and Medvedev, 2015). Tides are global-scale oscillations with periods of a solar day (24 h) and its harmonics (12 h, 8 h, 6 h), which mainly result from absorption of solar radiation by water vapor in the troposphere and

ozone in the stratosphere (Xu et al., 2012). Because of the decrease of density and conservation of energy, the tidal amplitudes increase with height (Chapman and Lindzen, 1970; Andrews et al., 1987) and reach a maximum in the MLT region before they dissipate. Tides with larger periods like diurnal tides (DTs), semidiurnal tides (SDTs) and terdiurnal tides (TDTs) usually have larger amplitudes than short-period tides like the quarterdiurnal tide (QDT). This is why in the past the QDT attracted less attention than the relatively well understood DT, SDT and TDT.

There are few observational and model studies on the QDT available. The QDT has been observed from satellites, so Azeem et al. (2016) analyzed temperature data from the Near-Infrared Spectrometer (NIRS) onboard the International Space Station (ISS) and from the Sounding of the Atmosphere using Broadband Emission Radiometry (SABER) instrument onboard the Thermosphere Ionosphere Mesosphere Energetics Dynamics (TIMED) satellite. They obtained an amplitude of the QDT which grew from $\sim 5\,\text{K}$ near $100\,\text{km}$ altitude to $\sim 30\,\text{K}$ near $130\,\text{km}$. Liu et al. (2015) also used measurements from SABER/TIMED

temperature data to analyze the QDT in the MLT and its global structure and seasonal variability. They compared the results with different Hough modes. In their study, they noticed that between 70 and 90 km altitude at the equator and low latitudes, the



(4,6) Hough mode dominated. Above 90 km, more than one Hough mode is visible, but the (4,6) mode remains predominant. The SABER/TIMED data show also a meridional structure with three amplitude maxima between $40°S$ and $40°N$; two of which are centered at $30°$ and one above the equator. This structure is also seen in the analyses of Azeem et al. (2016) with two additional maxima at about $60°$ on both hemispheres. Jacobi et al. (2019) analysed QDT signatures in lower ionospheric

sporadic E occurrence rates. They mainly found maxima during early and late winter at middle latitudes, which coincided with modeled vertical shear QDT maxima of the zonal wind.

    The QDT has been also observed in radar wind measurements in the MLT region (e.g., Sivjee and Walterscheid, 1994; Smith et al., 2004; Jacobi et al., 2017b; Guharay et al., 2018). Guharay et al. (2018) analyzed the variability of the QDT in the MLT over Brazilian low-latitude stations and found QDT wind amplitudes that reach $2\,ms^{-1}$ with a maximum during late summer

and fall. Jacobi et al. (2017b) analyzed MLT (80-100 km) radar data from Collm ($51°N$, $13°E$) and Obninsk ($55°N$, $37°E$). They found maximum amplitudes in winter with a long-term mean monthly mean zonal amplitude of $7\,ms^{-1}$. Bispectrum analysis of the Collm data showed that non-linear interaction is a possible forcing mechanism especially in winter and in the upper height gates accessible to the radar (Jacobi et al., 2018). MLT radar observations were also performed at Esrange ($68°N$, $21°E$) by Smith et al. (2004). They observed maximum monthly mean amplitudes during winter which can exceed $5\,ms^{-1}$.

They have also shown simulations of the QDT using a mechanistic model, which supports the results from the radar at Esrange and Collm with a similar timing and magnitude of the seasonal peak amplitude. Furthermore, Smith et al. (2004) analyzed the forcing mechanisms of the QDT and showed that the solar forcing mechanism is the most important one. Nevertheless, there is also a possible influence of a nonlinear interaction between different tides that may cause an additional QDT to the solar forced QDT. The theory of nonlinear interactions between tides was described by Teitelbaum and Vial (1991). Accordingly,

a pure nonlinear QDT wave (period of 6 h) is generated when a nonlinear interaction between two SDTs (periods of 12 h) or between a TDT and a DT (periods of 8 h and 24 h) occurs. Similarly, it holds for the wave numbers $k$, that a $k = 4$ wave can be formed by a nonlinear interaction between two existing waves with $k = 2$ or between waves with $k = 1$ and $k = 3$. Another possible source of tides is the interaction between gravity waves (GWs) and tides. For example, Miyahara and Forbes (1991) demonstrated such a mechanism for the TDT, but without consideration of the QDT. Simulations of GW-tide interactions were

performed by Ribstein and Achatz (2016), but they did not analyze higher harmonics than the SDT. Liu et al. (2006) showed nonlinear interactions between atmospheric tides at midlatitude radar measurements, as well as an interaction between tides and GWs from a bicoherence spectrum analysis. This was mainly found for the upper height gates considered.

    To summarize, there is some indication for the QDT as a regular phenomenon especially visible in the MLT, but the data base is sparse and there is no final and quantitative information about the role of its different forcing mechanisms. Therefore,

in this paper we analyze the migrating QDT in the middle atmosphere with the help of a mechanistic global circulation model, and focus on possible forcing mechanisms, i.e., the absorption of solar radiation by ozone and water vapor, nonlinear tidal interaction, and gravity wave-tide interaction. This will be done by separately analyzing these forcing mechanisms and their relative contribution to the QDT tidal amplitudes. The paper is structured as follows: At first the model and experiments are described, and the QDT model climatology is presented. After that the results of the runs with different forcing mechanisms

excluded are shown. Finally, the results will be discussed and summarized.





## 2   Description of the model and the experiments

The Middle and Upper Atmosphere Model (MUAM; Pogoreltsev, 2007; Pogoreltsev et al., 2007) is used to investigate the forcing mechanisms of migrating QDT with wave number 4. MUAM is a 3-D, primitive equation, mechanistic global circulation model based on the earlier COMMA-LIM model described by Jakobs et al. (1986), Fröhlich et al. (2003b) and Jacobi

et al. (2006). Recent versions of the MUAM model are described by Lilienthal et al. (2017, 2018), Lilienthal and Jacobi (2019), Jacobi et al. (2019) and Samtleben et al. (2019a, b). The model reaches from the surface at $1000\,hPa$ to $160\,km$ log-pressure height, with a constant scale height of $H = 7\,km$ and a vertical resolution of $2.842\,km$. In the lowermost $30\,km$, i.e. in the lowest 10 model levels, the zonal mean temperatures are nudged to monthly mean zonal mean ERA-Interim reanalysis temperatures (ERA-Interim, 2018; Dee et al., 2011). The wave propagation remains unaffected by nudging, as this only alters the

zonal mean. Above $30\,km$ the background winds can develop freely in the model, and are only affected by the zonal mean temperature nudging below. In contrast to other model experiments (e.g., Pogoreltsev et al., 2007; Pogoreltsev, 2007; Lilienthal et al., 2017; Samtleben et al., 2019a, b), here we do not include planetary wave forcing at the lower boundary to avoid undesired wave coupling with tides. In our experiments, we perform ensemble runs with 11 members using ERA-Interim data of the years 2000-2010, to describe interannual variability.

The solar heating through absorption, including water vapor, carbon dioxide, ozone, oxygen, and nitrogen, in the middle atmosphere is parameterized after Strobel (1978). Ozone is implemented as monthly mean zonal mean field for the year 2005 up to $50\,km$ altitude taken from MERRA-2 (Modern-Era Retrospective Analysis for Research and Application, version 2) reanalysis data (MERRA-2, 2019; Gelaro et al., 2017). Above $50\,km$, the ozone mixing ratio is assumed to decrease exponentially. In the ensemble runs, the ozone mixing ratio is chosen according to the Mauna Loa Observatory data for 2005 (e.g., 380

ppm for February 2005, NOAA ESRL Global Monitoring Division, 2018; Thoning et al., 1989), because we do not intend to perform an ozone dependent trend analysis. Extreme ultraviolet (EUV) and chemical heating (Riese et al., 1994) are included (see Fröhlich et al. (2003a)).

Tides are self-consistently forced in the model by the solar heating routines. The model is unable to produce non-migrating tides, because it contains no 3-D fields of ozone and water vapor, but only zonal means. In contrast to the version by Ermakova

et al. (2017) and Jacobi et al. (2017a), latent heat release is not included here. We used a horizontal resolution of $2.5° \times 5.625°$ for the model, which differs from the version of e.g., Lilienthal et al. (2017, 2018), to be able to better resolve the meridional structure of the QDT. In an earlier model version with $5°$ meridional resolution, essentially only one maximum in the QDT amplitudes per hemisphere was seen (Jacobi et al., 2019), while satellite observations (Azeem et al., 2016; Liu et al., 2015) show a more detailed meridional structure. Also from the linear theory including the QDT meridional structure representation

by Hough modes, another result was expected, as shown by Azeem et al. (2016).

The GW routine, which is used in this model version, is an updated Lindzen-type parameterization (Lindzen, 1981; Jakobs et al., 1986) as described by Fröhlich et al. (2003b) and Jacobi et al. (2006). This parameterization is based on waves initialized at $10\,km$ altitude, traveling in eight directions with phase speed between 5 and $30\,ms^{-1}$. These waves do not effectively propagate beyond the lowermost thermosphere, therefore the Lindzen-type routine is coupled with a modified parameterization





after Yiğit et al. (2008), initiated with GWs of higher horizontal phase speeds. The individually excited GWs are clearly separated through their different phase velocities. The distribution of tendency terms from both GW routines can be summed up to the total acceleration of the mean flow through GWs. More information about the GW parameterization included in MUAM is given in Lilienthal et al. (2018).

The model uses a time step of $120\,\mathrm{s}$ and starts with a spin-up time of $120$ model days. In that time the heating rates are zonally averaged, which means there are no tides. After that, further $90$ model days are simulated with zonally variable heating rates, so that there is tidal forcing now. The declination in this model version is fixed to the 15th day of the respective month. The following results that are presented are analyzed from the last $30$ model days. In this time period the tidal amplitudes remain almost constant and show only small day-to-day variations. Lower atmosphere mean temperatures are nudged during

the entire model run. However, since only zonal means are modified, tidal forcing and propagation remains possible.

Solar tides, including the QDT, may be generated by three different mechanisms, named solar heating, nonlinear tide-tide interactions, and GW-tidal interactions. More details of these forcing mechanisms and how they are represented in the MUAM model are described by Lilienthal et al. (2018). Here, we essentially follow their approach by removing different forcing mechanisms. To this end, we used a Fourier transform and removed the wave number $4$ (which is equivalent to the migrating QDT,

since there are no non-zonal structures except for the migrating tides in our MUAM version) amplitude from the respective forcing term during each time step and at each model grid point. To remove the solar forcing mechanism, the wave number $4$ heating was removed from the radiation parameterization scheme. To remove the nonlinear tide-tide interactions, we separated the nonlinear terms, which are essentially the advection terms in the momentum equation and the temperature equation as has been done in Lilienthal et al. (2018). Then we removed the wave number $4$ in these terms. Since these advection terms are

responsible for wave-wave interaction, this strategy effectively removes the QDT forcing through non-linear interaction. To remove GW-tidal interaction, the total acceleration and heating through GW oscillations of wave number $4$ are removed. Table 1 shows an overview of our simulations, in which different forcing mechanisms are eliminated separately: (i) SOL with no GW-tidal interactions and no nonlinear interactions, (ii) NLIN, without solar forcing mechanism and without GW-tidal interactions, and (iii) GW without solar forcing and without nonlinear interactions. Effectively, these experiments represent model runs with

only solar (SOL), nonlinear (NLIN), and GW (GW) forcing of the QDT. Furthermore, two experiments were performed where only one process was removed, namely (iv) NO_NLIN with removed nonlinear interactions, and (v) NO_GW without GW-tidal interaction. In addition, a reference (REF) run was performed with all forcing mechanisms enabled.

## 3   Results

### 3.1   Reference simulation and QDT climatology

In the reference run (REF), all forcing mechanisms (direct solar, GW-tide interactions and nonlinear interactions) are included. Results from this experiment will be described here in comparison with results from the literature. For an overview of the seasonal cycle of the QDT, Fig. 1 shows the QDT temperature and wind amplitudes at about $101\,\mathrm{km}$ height. The amplitudes





maximize in autumn and winter at higher midlatitudes of both hemispheres with two maxima formed between 20°-40° and 50°-70°, respectively. In the northern hemisphere, largest amplitudes are found in February and October.

Liu et al. (2015) showed a climatology of QDT temperature amplitudes from SABER/TIMED satellite data between 50° North and South over 10 years. The amplitudes presented by Liu et al. (2015) show maxima near 30° North and South and

above the equator. Their QDT temperature amplitudes reach values of 0.5 K to 1.0 K between 70 km and 90 km, and at higher altitudes the amplitudes reach up to 4 K on an annual and long-term average. Thus, the amplitudes observed by Liu et al. (2015) are larger than in the MUAM simulation.

The maxima in February, April, May and August at 40°N from MUAM simulations in Fig. 1 (a) agree with the satellite measurements analyzed by Liu et al. (2015). Our simulated maximum in October, on the other hand, does not appear in the

SABER/TIMED data. Also, the extrema at about 10°N in June, September and October from Liu et al. (2015) do not match with the MUAM results. The largest QDT amplitudes in the southern midlatitudes derived from the satellite data do not show agreement with the MUAM results in Fig. 1 (a).

Model simulations of the QDT temperature amplitudes at 100 km altitude by Smith et al. (2004) show a similar seasonal and latitudinal amplitude maximum distribution as seen in the MUAM results. Again, however, the amplitudes in the model

simulations from Smith et al. (2004) are larger than in the MUAM results. Amplitudes in the MUAM simulations tend to underestimate other results by a factor of about 2 or 3. One reason for this is that water vapor in MUAM is implemented as zonal mean and not as 3-D field and that latent heat is not included as a QDT source in the model. In addition, the amplitudes of other tides (DT, SDT, TDT) are also too small compared to observations (Lilienthal et al., 2018), so that nonlinear interaction processes are possibly underestimated.

Meteor radar measurements of zonal wind QDT amplitudes at 50°N by Jacobi et al. (2017b, 2018) show maxima in January and February, as well as in April and May, analogous to the MUAM simulations. The maxima in autumn seen in Fig. 1 b are also supported by their measurements. Also, the temporal and spatial distribution of zonal wind amplitudes bySmith et al. (2004) show good similarity in with the MUAM simulations. The same is the case for the meridional wind amplitudes in Fig. 1 c.

Ensemble simulations, which contain the solar, nonlinear and GW forcing mechanism for all wave numbers, are useful as a reference for all experiments because they represent a QDT that can be compared with observations. The results in the following are given as means of the 11 ensemble members. Since in the northern hemisphere the largest amplitudes are found in February and October (see Fig. 1) we selected these months for further analysis. In Fig. 2 the background climatology for the MUAM zonal mean circulation is shown for February (a, b) and October (c, d), for the parameters temperature (a, c) and

zonal wind (b, d). The data are the model results for the years 2000—2010 (color coding), i.e. with the respective ERA-Interim reanalysis zonal mean temperatures used for nudging, with the corresponding standard deviations (contour lines).

The model zonal wind climatology agrees reasonably well with earlier empirical climatologies such as CIRA86 (Fleming et al., 1990) or the radar-based GEWM (Portnyagin et al., 2004; Jacobi et al., 2009) and the satellite-based URAP (Swinbank and Ortland, 2003). In February the easterly jet of the summer hemisphere is weaker in comparison with the climatologies. The

same is true for the equatorial easterly winds in October. The model temperature shows general agreement with the empirical





CIRA86 climatology. In February the stratopause and mesopause temperatures above the equator and low latitudes are about $10\,\mathrm{K}$ lower than predicted by the CIRA86 climatology. These differences are not seen in the comparison for October. MUAM produces a year-to-year variability (standard deviation $\sigma$) especially in the areas of the strongest jets of the northern mid-latitudes in February (up to $\sigma(u) = 8\,\mathrm{ms}^{-1}$) and at the southern midlatitudes in October (up to $\sigma(u) = 10\,\mathrm{ms}^{-1}$). The reason

for this is the annual variability in the formation of the polar vortex, which affects the strength of the jets and the temperature at the high and midlatitudes. This variability causes fluctuation of a few K or $\mathrm{ms}^{-1}$. Elsewhere, the standard deviation is very small, and mostly amounts to less than $\sigma(T) = 2\,\mathrm{K}$ ($\sigma(u) = 2\,\mathrm{ms}^{-1}$, $\sigma(v) = 0.5\,\mathrm{ms}^{-1}$).

In comparison with the more recent Horizontal Wind Model (HWM14, Drob et al., 2015), the westerly wind jet in February in the middle atmosphere midlatitudes is much stronger ($+20\,\mathrm{ms}^{-1}$) in the MUAM simulation. The easterly wind jet in the

mesosphere, on the other hand, is much weaker ($-35\,\mathrm{ms}^{-1}$) in the MUAM simulation than predicted HWM14. Also, the mesospheric wind reversal found at higher altitudes in HWM14 ($100\,\mathrm{km}$) than in the MUAM ($80\,\mathrm{km}$) simulation, especially in the northern hemisphere. Similarly, the wind jets in the mesopause and lower thermosphere region are much weaker in the MUAM run than in HWM14. A better agreement is seen for October regarding the strength of the wind jets. However, in contrast to February, the wind reversal in October is higher in MUAM ($80\,\mathrm{km}$) than in HWM14 ($70\,\mathrm{km}$).

All QDT forcing terms, including the solar forcing, nonlinear forcing and the forcing resulting from GW-tide interactions, are shown in Fig. 3 (thermal parameters) and Fig. 4 (wind parameters) for February (left panels) and October (right panels). All these forcing terms in the MUAM tendency equations are scaled by the factor $exp[-z(2H)^{-1}]$ in order to account for the growth rate of the amplitudes with altitude due to decreasing density. Thus, the figures show the source regions of the QDT. However, from the parameters shown in Fig. 3 and 4 no statement about the propagation conditions of the QDT is possible

because the tide might be trapped in the source region, not being able to propagate upwards (Lilienthal et al., 2018). In general, the QDT in-situ forcing in February and October shows a similar global distribution.

Figure 3 shows temperature advection (a, b), the nonlinear component of adiabatic heating (c, d), the heating related to dissipating GWs (e, f), and direct solar heating (g, h). Note the different color scales in Fig. 3 to cover the maxima of all forcing terms. The thermal forcing (Fig. 3) of the QDT is dominated by direct solar heating in the troposphere and stratosphere

(g, h). This is due to the absorption of solar radiation by water vapor in the troposphere and ozone in the stratosphere. In the mesosphere (80-110 km) nonlinear wave-wave interactions (a, b) play the most important role and show maxima at the equator in the stratosphere, mesosphere and lower thermosphere. Nonlinear adiabatic heating (c, d) maximizes in the upper stratosphere and mesosphere at the equator. However, this forcing is about one order of magnitude smaller than the nonlinear forcing and therefore will be disregarded in the following. In the lower thermosphere, the strongest QDT generation second

to solar heating takes place through GW heating (e, f). Nevertheless, nonlinear effects continue to occur, and they are partly comparable in magnitude with the GW forcing.

Figure 4 shows QDT acceleration terms in the momentum equations, and thus refers to the wind parameters. The data are again scaled by $exp[-z(2H)^{-1}]$ according to energy conservation. The different panels show the zonal (a, b) and meridional wind advection (c, d) as well as the zonal (e, f) and meridional (g, h) acceleration due to GWs. In the troposphere, stratosphere

and large parts of the mesosphere, the nonlinear forcing of both the zonal (a, b) and meridional (c, d) QDT wind dominates over





the GW forcing (e - h). Near the mesopause, GW zonal and meridional forcing is more important than the nonlinear forcing in zonal and meridional wind. The zonal GW forcing becomes relatively strong above 60 km at the northern middle latitudes. The GW forcing plays a major role above 110 km, where it dominates over other nonlinear forcings. In the meridional component, the wind advection (c, d) outweighs the GW forcing (g, h) at almost all altitudes.

## 3.2 Separation of quarterdiurnal generation mechanisms

To quantify the effect of each forcing mechanism on the QDT, we performed simulations with various forcing terms switched off (see Table 1). For the months of February and October, the QDT amplitudes and phases of the simulations REF, SOL, GW and NLIN are shown in Figs. 5 – 8 (Fig. 5 a, b temperature and Fig. 7 a, b zonal wind). Note that amplitudes are not scaled in contrast to the forcing terms in Fig. 3 and 4. In October the amplitudes tend to be a little stronger than in February Fig. 5, and generally the amplitudes increase with height. In the REF run, there are four maxima for temperature and zonal wind. At 100 km altitude, amplitudes up to $0.5$ K in temperature and $1.5$ ms$^{-1}$ in zonal wind are achieved. Thus, the modeled amplitudes are much smaller than reported from measurements (e.g., Liu et al., 2015; Azeem et al., 2016; Jacobi et al., 2017b; Guharay et al., 2018), i.e. satellite measurements reveal temperature amplitudes of 5-10 K, depending on season and altitude, while radar data suggest wind amplitudes of 2.5-5 ms$^{-1}$.

In Fig. 5 c, d the SOL simulations for February and October are shown for temperature and in Fig. 7 c, d for zonal wind. In this run, the GW forcing mechanism and the nonlinear forcing mechanism have been removed from the terms of the model tendency equation as described in section 2. The QDT amplitudes in the SOL run look very similar to those of the REF run in terms of amplitude magnitude and distribution. This agrees well with Fig. 3 g, h, showing that direct solar forcing is the strongest forcing mechanism and dominates the QDT in-situ generation. On closer examination, the midlatitudes of southern and northern hemisphere show even larger temperature and zonal wind amplitudes in the SOL run than in the REF run, in particular during February. On the other hand, amplitudes during October tend to be slightly decreased in the SOL simulation but with similar global structure like those of the REF simulation. The GW run only contains the GW forcing and shows only small amplitudes for the temperature (Fig. 5 e, f, up to 2 K) and zonal wind (Fig. 7 e, f, up to $3.5$ ms$^{-1}$) compared to the REF and SOL simulations. Similar to the REF simulation, amplitudes gradually increase with height and maxima are located at northern low latitudes of the lower thermosphere, however, they are negligible below 115 km. This is most likely due to the fact that GW-tide interactions mainly take effect in the lower thermosphere (see Fig. 3 and 4).

Figures 5 g, h and 7 g, h show the QDT amplitudes for the NLIN run. This simulation contains only the forcing of nonlinear interactions.The amplitudes for the temperature component (Fig. 5 g, h) are comparable to those of the GW run with a maximum of 2 K. For the zonal wind component (Fig. 7 g, h the amplitudes are even smaller than in the GW run with less than $1.5$ ms$^{-1}$. Therefore, we cannot derive a clear meridional structure of the nonlinear QDT. Keeping in mind that nonlinear tidal interactions mainly occur in the mesosphere (see Fig. 3 and 4), one may conclude that QDTs generated by this mechanism are trapped near their forcing region and cannot propagate further upward.

In addition, a NO_NLIN (NO_GW) run was performed in which only quarterdiurnal nonlinear interactions (GW-tide interactions) have been removed. The amplitudes (Fig. S1 and S3) and phases (Fig. S2 and S4) of these simulations are shown in the





supplement (Figs. respectively), because amplitude and phase differences compared to the REF simulation are rather small. However, similar to the SOL simulation, the amplitudes of NO_NLIN (NO_GW) are partly even larger than in REF. Similar behavior has been reported by Smith et al. (2004), who removed the nonlinear QDT forcing in their model and concluded that tidal interactions rather reduce than enhance the QDT amplitude. In the following, this will be investigated in more detail

by analyzing phase differences between the differently generated QDTs. This way, we intend to reveal possible interactions between these waves.

The corresponding phases of the REF simulation can be found in Fig. 6 a, b for temperature and in Fig. 8 a, b for zonal wind. The corresponding vertical wavelength can be determined at any latitude from the vertical phase gradient. The wavelength is defined by the vertical distance between two points with identical phases and should cover a complete span of phases.

According to theory, an upward propagating wave must have a negative phase gradient. At latitudes with large amplitudes, the vertical wavelengths tend to be larger, as well. In the opposite case, the wavelengths are smaller when the amplitudes are small. In February the wavelengths reach 100 km and more. In October phases are very similar. Both months show large areas with constant phases, especially at low latitudes.

Also, the QDT phases for the temperature (Fig. 6 c, d) and zonal wind (Fig. 8 c, d) component of the SOL simulation are

almost identical with the results of the REF run. The phases of the GW run (Fig. 6 d, f and 8 d, f) clearly differ from the REF run, i.e. vertical wavelengths are shorter and the phase position and distribution have also changed. Looking at the QDT phases of the NLIN run for temperature (Fig. 6 g, h) and zonal wind (Fig. 8 g,h), the associated vertical wavelengths are again smaller compared to the GW run, based on a more irregular phase distribution.

In Fig. 9, we present amplitude differences of the QDT between the NO_NLIN and REF simulation (color coding) where

red (blue) areas denote larger amplitudes in NO_NLIN (REF); in other words, the amplitude increases (decreases) when the nonlinear QDT forcing mechanism is removed. Note that these amplitude differences are scaled by the growth rate of the tides with altitude to highlight the actual source region of the waves. Figure 9 a, c shows the temperature component and (b, d) the zonal wind component in February and October, respectively. Furthermore, the hatched areas denote destructive interference between the QDTs of NLIN and SOL, which are defined through their phases differences $\Delta\Phi = \Phi_{NLIN} - \Phi_{SOL}$:

$$120° \leq \Delta\Phi \leq 240°. \tag{1}$$

In case of a superposition of such destructively related NLIN and SOL waves, the amplitude of NO_NLIN is expected to be larger than in REF, because the nonlinear (NLIN) and solar (SOL) QDT of the REF run act against each other. Indeed, we observe regions for temperature (Fig. 9 a, c) and zonal wind (Fig. 9 b, d) in which the amplitudes in the NO_NLIN run are larger than in the REF simulation, and at the same time, destructive interference between the nonlinear and solar QDT corresponds

to these positive amplitude differences. Thus, we can conclude that the nonlinearly excited part of the QDT weakens the pure solar QDT amplitude in the REF simulation. The effect is more pronounced for the zonal wind than for temperature.

In addition to the interaction between nonlinear and solar QDT, an interaction between GW-induced QDT and solar QDT is also possible. For this reason we show the respective results in Fig. 10, analogue to Fig. 9. Colors denote the differences between the NO_GW and the REF simulation, again scaled by the growth rate of the amplitudes with altitude. Red (blue) colors





denote larger `NO_GW` (`REF`) amplitudes. Areas of destructive interference (see Eq. 1 with $\Delta\Phi = \Phi_{GW} - \Phi_{SOL}$) between the phases from the `GW` and `SOL` run are hatched. The difference between `NO_GW` and `REF` run shows that the amplitudes in the `NO_GW` simulation are sometimes larger than in the `REF` run. This often happens in areas where destructive interference can be detected, but it is less well pronounced than in Fig. 9 for the nonlinear-solar QDT interaction. This means that the QDT

owing to GW-tide interactions also tends to act against the solar QDT which leads to a decline in QDT amplitude in the `REF` simulation for temperature and zonal wind where both forcing mechanisms are present. The interaction between `GW` and `NLIN` QDT is not shown separately because they turn out to be negligible.

## 4  Discussion and conclusion

The results of the `REF` simulation show a consensus in the climatology and global structure of QDT in comparison with

observations and other model studies. The amplitudes of the MUAM are relatively small for the QDT with up to $2.5\,\mathrm{ms}^{-1}$ in the zonal wind at $101\,\mathrm{km}$ altitude and $5.0\,\mathrm{ms}^{-1}$ at $120\,\mathrm{km}$ altitude in spring and autumn. For example, QDT amplitudes obtained from meteor radar measurements (Jacobi et al., 2017b) are up to three times larger than in the MUAM simulations. However, it is a known issue that numerical models tend to underestimate the tides in some regions and seasons (e.g., Smith, 2012; Pokhotelov et al., 2018).

In our simulations, the meridional structure of QDT amplitudes shows 3-4 maxima in both the temperature and zonal wind component. These are located at low ($10°$-$30°$) and middle latitudes ($40°$-$70°$) of the respective hemisphere. These maxima at low and midlatitudes are also present in the NIRS and SABER temperature measurements (Liu et al., 2015; Azeem et al., 2016). Meteor radar measurements at northern midlatitudes (Jacobi et al., 2017b) confirm our QDT wind maxima in winter, spring and autumn. The maximum of the QDT wind amplitudes at low latitudes has been proven by meteor radar measurements

over Brazil (Guharay et al., 2018). They show maxima below $100\,\mathrm{km}$ in spring and autumn like the MUAM simulations.

In the present paper we focused on forcing mechanisms of the QDT. To this end, we first compared all possible sources of the migrating QDTs in our simulations following the approach of Lilienthal et al. (2018). These are (i) the absorption of solar radiation by water vapor and ozone, (ii) nonlinear tidal interactions between migrating DTs and TDTs and the self-interaction of migrating SDTs and (iii) nonlinear interactions between GWs and tides. To our knowledge, this is the first time to present

the global distribution of quarterdiurnal in-situ forcing from a numerical model. In summary, the solar forcing dominates in the troposphere and stratosphere, the nonlinear forcing predominates in the mesosphere and the GW forcing mainly takes place in the mesosphere and thermosphere. These results do not allow to draw conclusions on the upward propagation of the QDT, but only show local excitation.

For this reason, we adapt the idea of Smith et al. (2004), who performed simulations with individual forcing mechanisms

removed. In addition to Smith et al. (2004), we also consider GW-tide interactions. Some of our simulations are designed in a way that only a single forcing mechanism remains and the other two sources are removed (`SOL`, `NLIN` and `GW`), in other simulations only one of the sources was removed (`NO_NLIN`, `NO_GW`).





As a result, we find that the solar forcing mechanism is the most important and dominant one of all forcing mechanisms, since the removal of direct quarterdirunal solar heating (GW and NLIN runs) leads to a significant decrease in the QDT amplitude. Smith et al. (2004) came to the same conclusion, when they removed the quarterdiurnal solar forcing in their simulations.

We also showed that the amplitudes resulting from the GW forcing mechanism (GW) are smaller than the resulting amplitudes
of the direct solar forcing (SOL), but larger than those from the nonlinear forcing mechanisms (NLIN). In agreement with the results of Smith et al. (2004), nonlinear tidal interactions seem to play a minor role for the total QDT amplitudes, although we found distinct sources of nonlinear quarterdiurnal in-situ excitation in the mesosphere (see above). This allows the conclusion that the QDT from local nonlinear forcing mechanisms can not propagate and is, to a large degree, trapped in the vertical domain. Significant nonlinear QDT amplitudes only exist in the thermosphere. In the temperature component, QDT amplitudes
of the NLIN and GW simulation are comparable in magnitude. In the zonal wind component, they are smaller in NLIN than in GW. For the GW and NLIN simulations we note relatively short vertical wavelengths, accompanied by small QDT amplitudes, compared to the SOL and REF runs. So we can state that if the amplitudes are small, the vertical wavelength is shorter as well. Lilienthal et al. (2018) has found a similar relation for the vertical wavelengths of the TDT.

In the SOL simulation, which only contains the solar forcing, we see that the amplitudes are in some cases larger than in
the REF run. A similar feature has been observed by Smith et al. (2004). Here, we compare phase and amplitude differences between our different simulations to investigate the physical explanation behind. We find that the amplitudes in simulations with removed forcing mechanisms (NO_NLIN and NO_GW) increase compared to REF in the same areas where destructive phase relations between the differently generated QDTs are detected. This leads to the conclusion that QDTs excited by different mechanisms counteract rather than enhance each other. Thus, removing an individual forcing mechanism in NO_NLIN or
NO_GW also avoids the destructive interference and the remaining QDT can propagate freely, resulting in larger amplitudes.

This destructive relation appears to be more clear between the nonlinear tidal forcing and the direct solar forcing than between the GW-induced forcing and the solar forcing. Note, however, that nonlinear tidal interactions generally have a smaller impact on the QDT than GW-tide interactions, as described above. We did not present phase relations between the nonlinear and GW forcing because these turned out to be small. Apparently, the dominating solar forcing has to be involved in the de-
structive phase relation. In future, an implementation of a latent heat release parameterization according to Ermakova et al. (2019) and Jacobi et al. (2017a) and three-dimensional ozone (Suvorova and Pogoreltsev, 2011) and water vapor (Ermakova et al., 2017) fields into the model is planed, which may help to increase tidal amplitudes towards more realistic magnitudes. Another important issue is the careful treatment of GWs, because we demonstrated that GWs are the most important source of QDTs above the mesopause. In MUAM, GWs are currently implemented via two coupled parameterizations. These two
parameterizations could be replaced by the original whole atmosphere scheme, such as provided by Yiğit et al. (2008). Furthermore, a sensitivity study with respect to the strength of the individual forcing terms may contribute to a better understanding of the forcing mechanisms and interactions. Thereby, we intend to show their impact on QDT amplitudes and the background circulation. Further examination of dominating Hough modes may help explain the different meridional structures at different altitudes.



*Code availability.*  The MUAM model code can be obtained from the corresponding author on request.

*Author contributions.*  CG performed and designed the MUAM model runs together with FL. CG drafted the first version of the text. Analysis and interpretation of the results were contributed by CJ and FL.

*Competing interests.*  C. Jacobi is one of the Editors-in-Chief of Annales Geophysicae. The authors declare that no competing interests are
5  present.

*Acknowledgements.*  This research has been funded by Deutsche Forschungsgemeinschaft under grant JA 836/34-1. MERRA-2 global ozone fields were provides by NASA through https://disc.gsfc.nasa.gov/datasets?keywords=%22MERRA-2%22&page=1&source=Models%2% FAnalyses%20MERRA-2 (MERRA-2, 2019; Gelaro et al., 2017). Mauna Loa carbon dioxide mixing ratios were provided by NOAA through ftp://aftp.cmdl.noaa.gov/data/trace_gases/co2/flask/surface/ (Thoning et al., 1989; NOAA ESRL Global Monitoring Division, 2018). ERA-
10  Interim data have been provided by ECMWF on https://apps.ecmwf.int/datasets/data/interim-full-moda/levtype=sfc/ (Dee et al., 2011; ERA-Interim, 2018).



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





**Table 1.** Overview on the different modell experiments

| Simulation | Description | Solar forcing | Nonlinear forcing | Gravity wave forcing |
|---|---|---|---|---|
| SOL | Removed nonlinear and GW forcing | on | off | off |
| NLIN | Removed solar and GW forcing | off | on | off |
| GW | Removed solar and nonlinear forcing | off | off | on |
| NO_NLIN | Removed nonlinear forcing | on | off | on |
| NO_GW | Removed GW forcing | on | on | off |
| REF | Reference with all forcings | on | on | on |

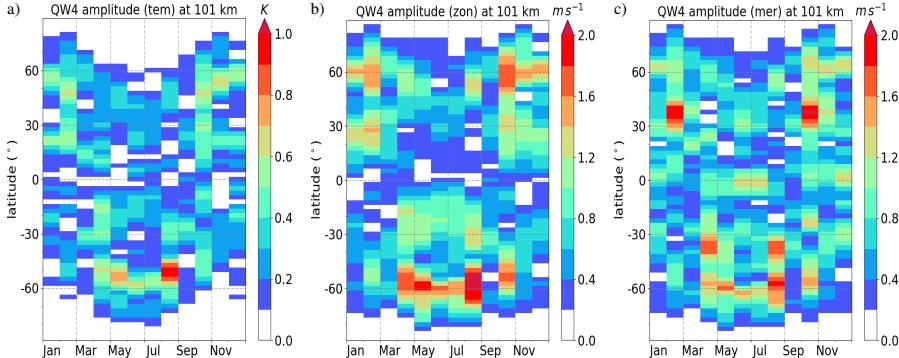

**Figure 1.** REF monthly mean QDT amplitudes at 101 km altitude. From left to right: (a) temperature, (b) zonal wind, (c) meridional wind.



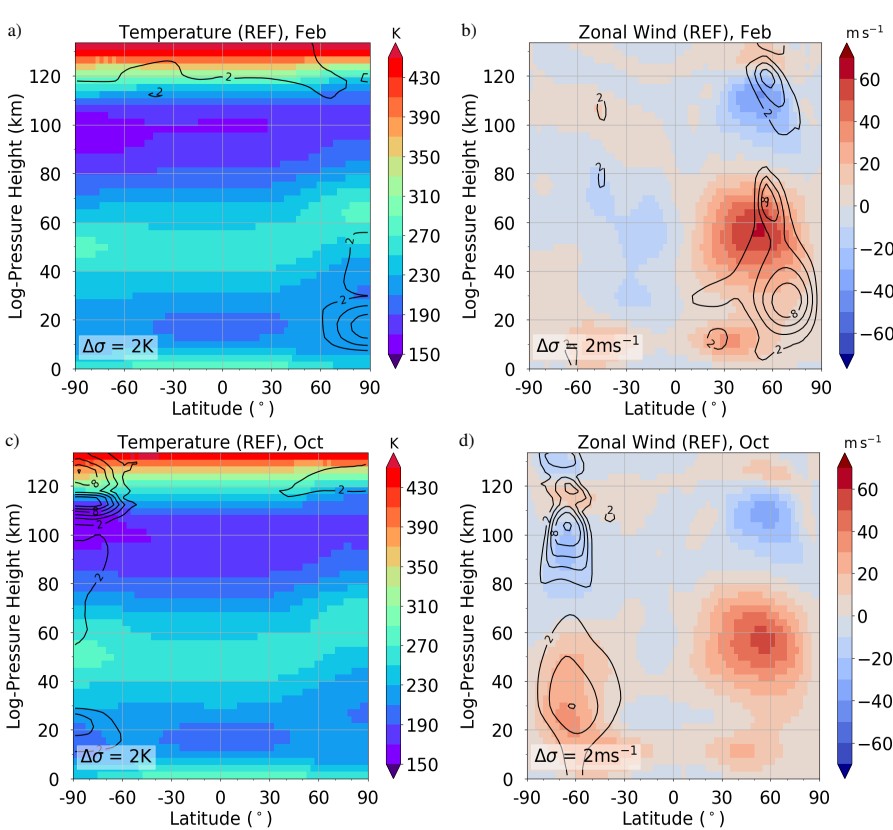

**Figure 2.** (a,c) `REF` zonal mean temperature and (b,d) zonal wind. (a-b) February conditions. (c-d) October conditions. Results are an average of 11 ensemble members (shaded color). Standard deviations $\sigma$ are $2\,\mathrm{K}$ for temperature and $2\,\mathrm{ms}^{-1}$ for zonal wind.



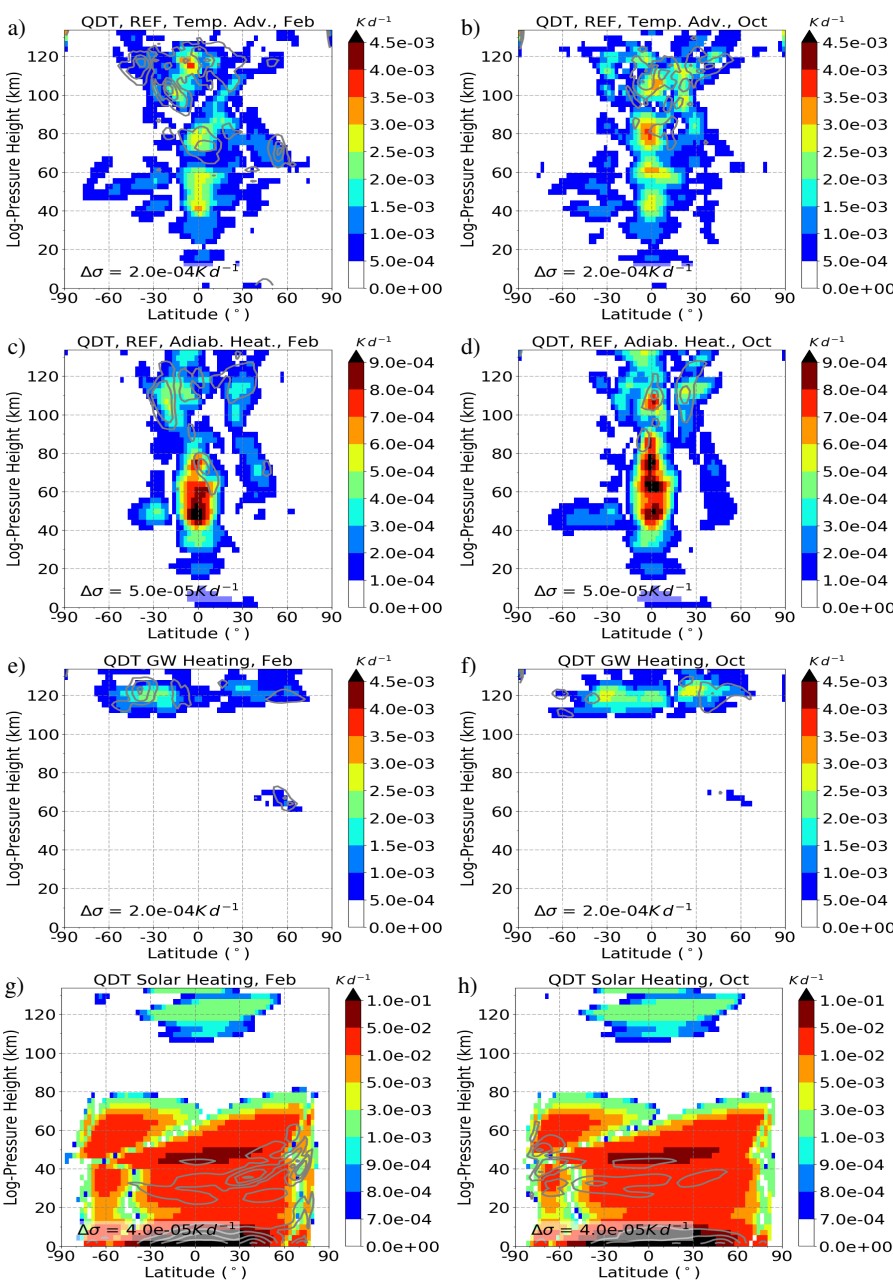

**Figure 3.** Quarterdiurnal component of thermal tendency terms in the REF simulation for February conditions (a, c, e, g) and October conditions (b, d, f, h). Amplitudes are scaled by $exp[-z(2H)^{-1}]$. Results are an average of the 11 ensemble members (shaded color). Standard deviations ($\sigma$) are added as grey contour lines. (a, b) Temperature advection (nonlinear component), (c, d) adiabatic heating (nonlinear component), (e, f) heating due to GW activity (tendency term from GW parameterization), and (g, h) solar heating (tendency term from solar radiation parameterization). Note that the color scales are different, and that the scale in panels (g, h) is not linear.



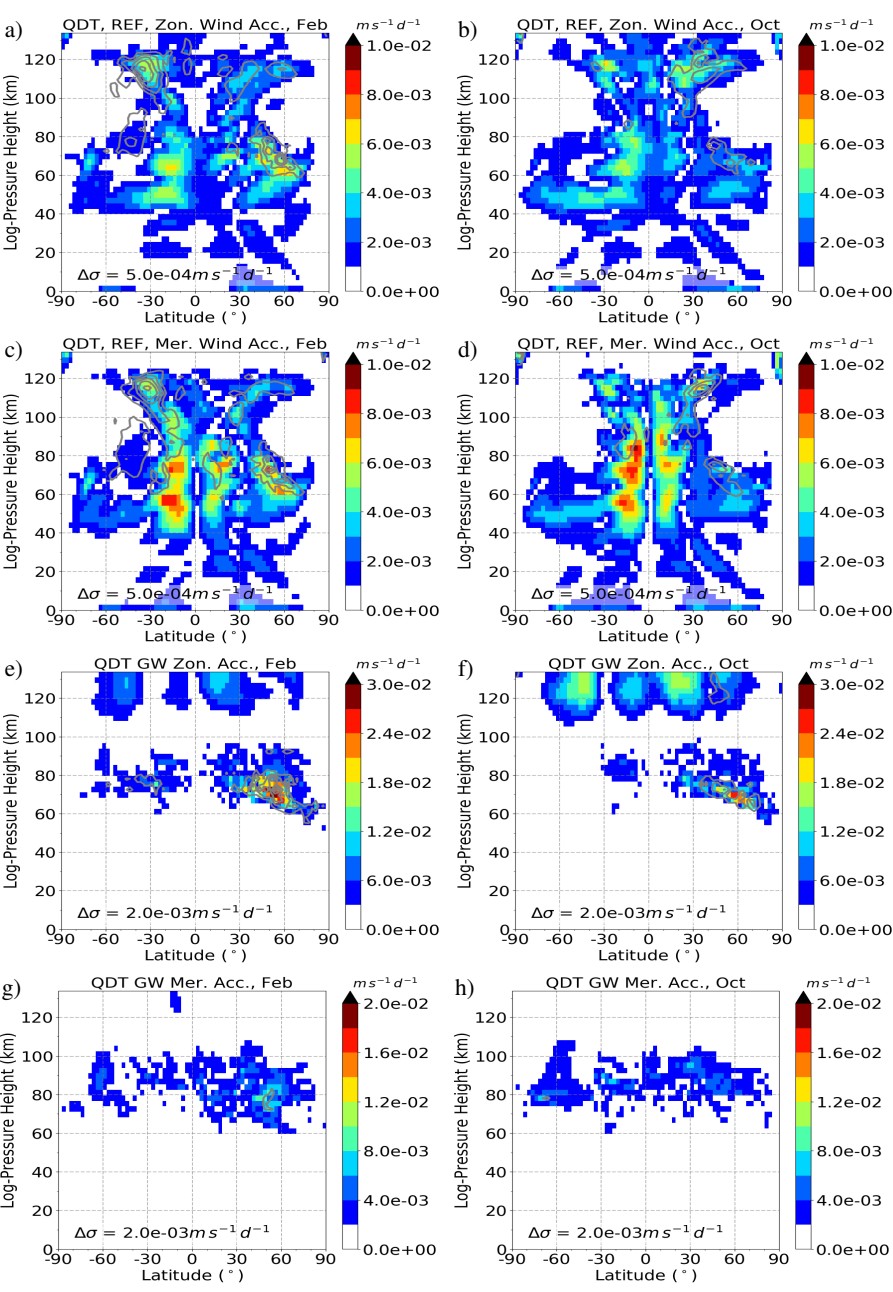

**Figure 4.** Quarterdiurnal component of zonal and meridional wind acceleration terms in the REF simulation for February conditions (a, c, e, g) and October conditions (b, d, f, h). Amplitudes are scaled by $exp[-z(2H)^{-1}]$. Results are an average of the 11 ensemble members (shaded color). Standard deviations ($\sigma$) are added as grey contour lines. (a, b) zonal wind advection (nonlinear component), (c, d) meridional wind advection (nonlinear component), (e, f) zonal and (g, h) meridional acceleration due to GWs (tendency terms from GW parameterization). Note that the color scales are different.



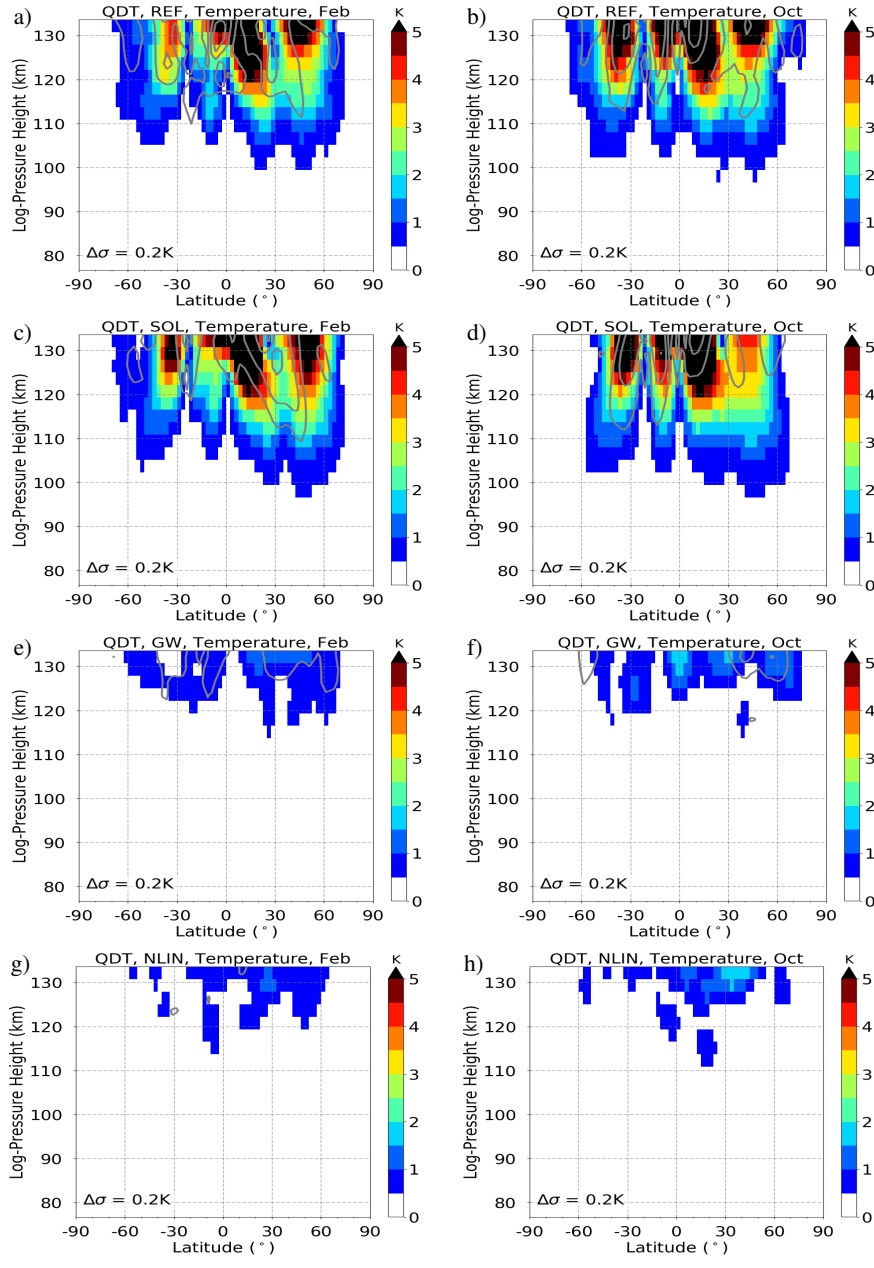

**Figure 5.** Simulations of zonal mean QDT amplitudes for temperature (colors). (a, c, e, g) February and (b, d, f, h) October conditions. (a, b) REF run with all forcing mechanisms enabled, (c, d) SOL run with just direct solar forcing mechanism enabled, (e, f) GW run with just GW forcing mechanism enabled and (g, h) NLIN run with just nonlinear forcing mechanism enabled. Standard deviations $\sigma$ are added as gray contour lines.



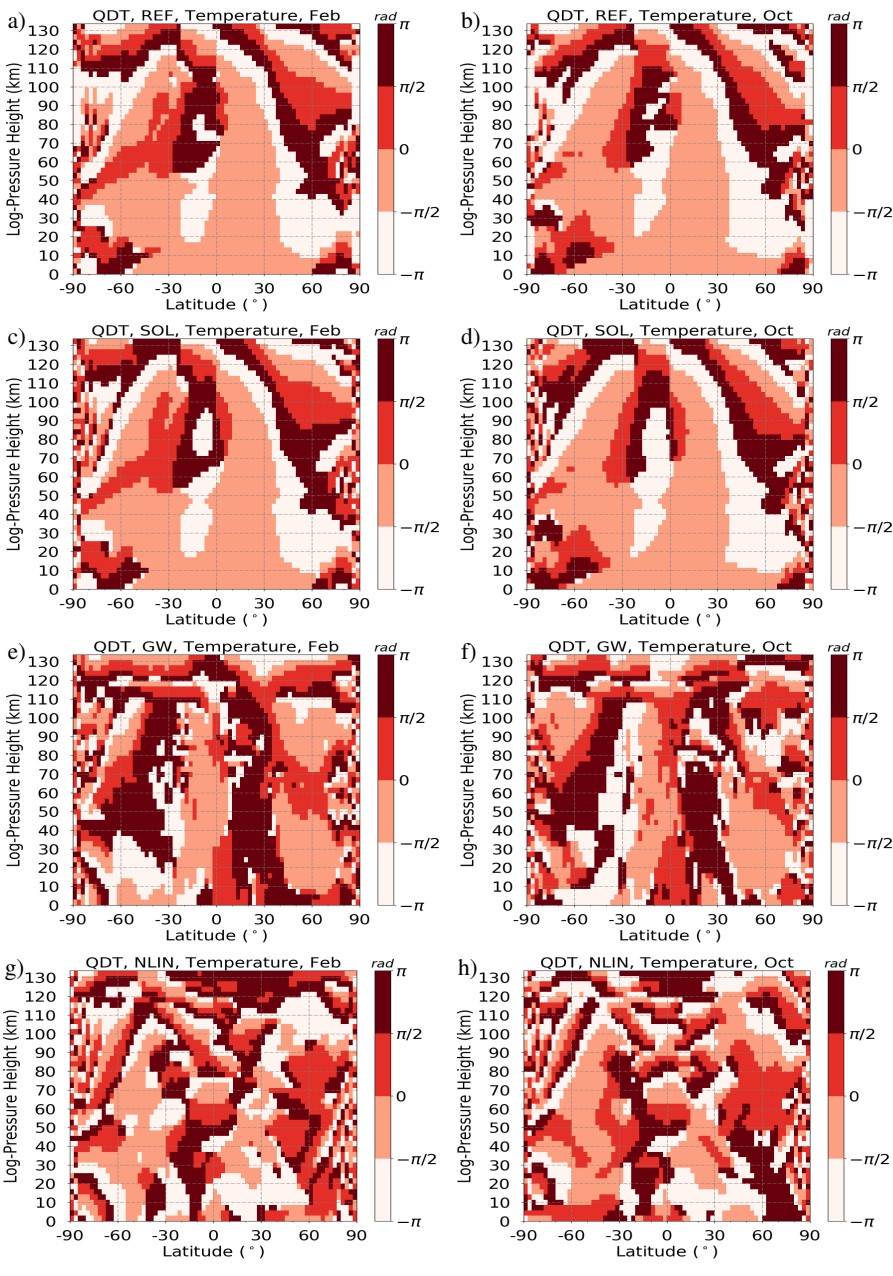

**Figure 6.** Simulations of zonal mean QDT phases for temperature (colors). (a, c, e, g) February and (b, d, f, h) October conditions. (a, b) REF run with all forcing mechanisms enabled, (c, d) SOL run with just direct solar forcing mechanism enabled, (e, f) GW run with just GW forcing mechanism enabled and (g, h) NLIN run with just nonlinear forcing mechanism enabled. Standard deviations $\sigma$ are added as gray contour lines.





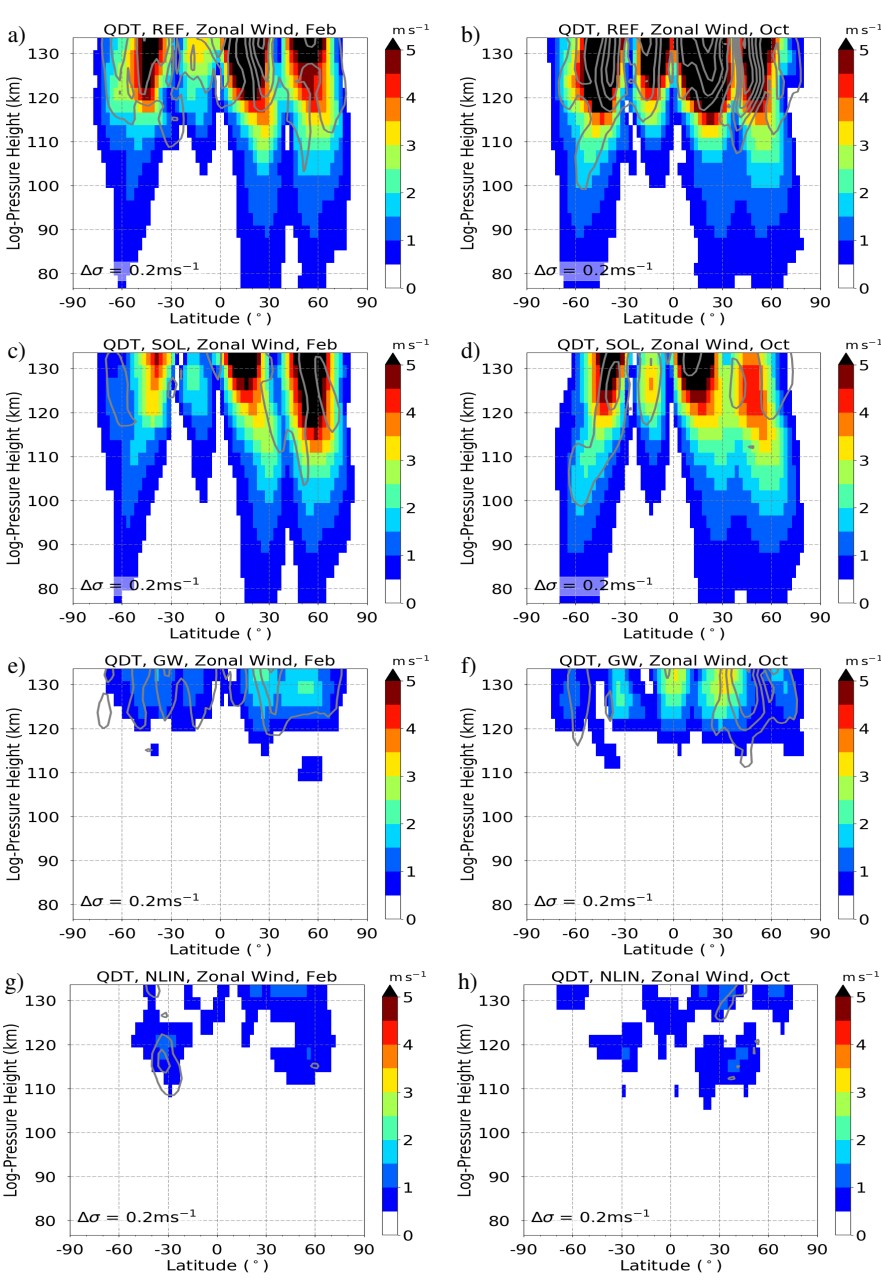

**Figure 7.** Same as Fig. 5 but for QDT zonal wind amplitudes.



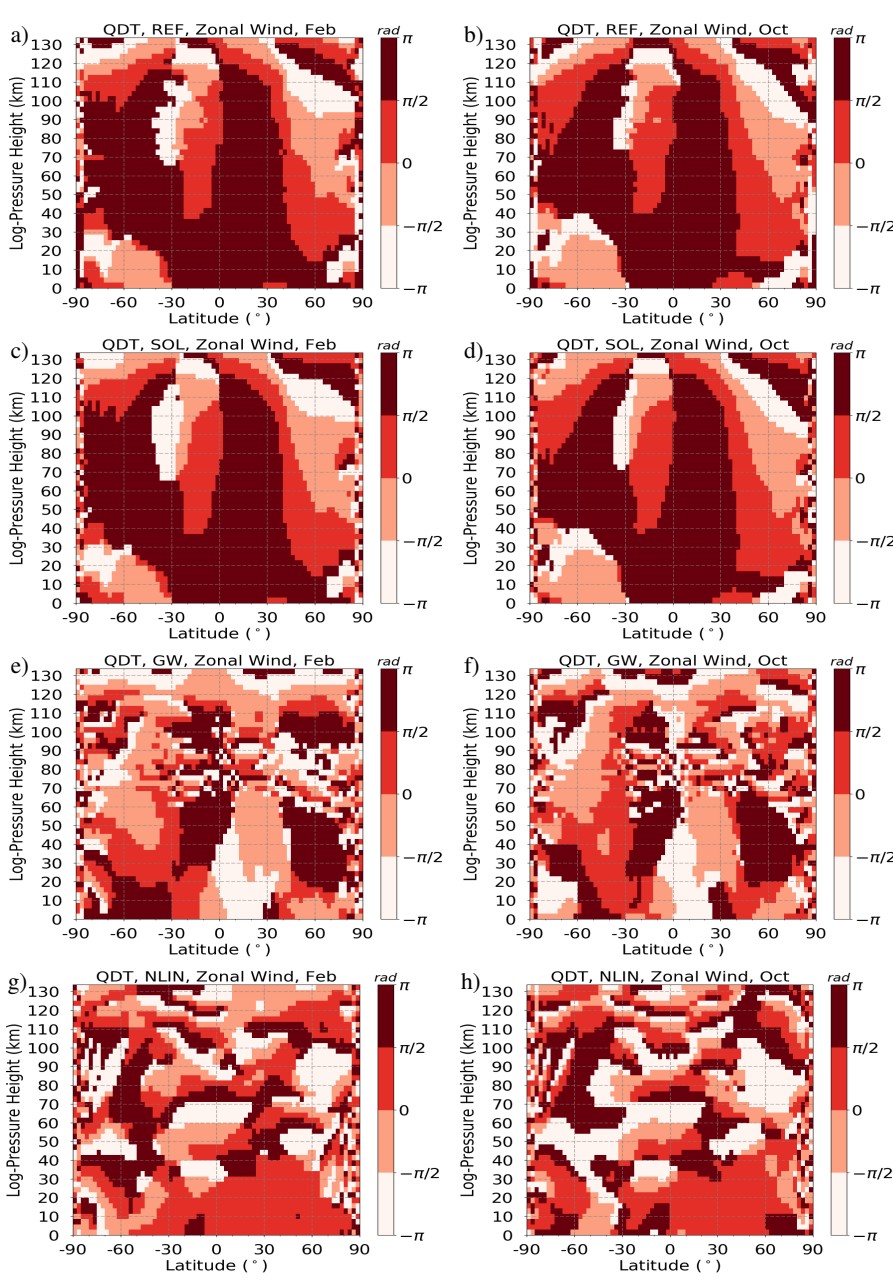

**Figure 8.** Same as Fig. 6 but for QDT zonal wind phases.



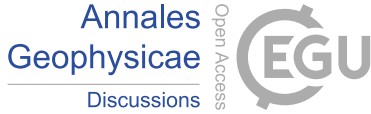

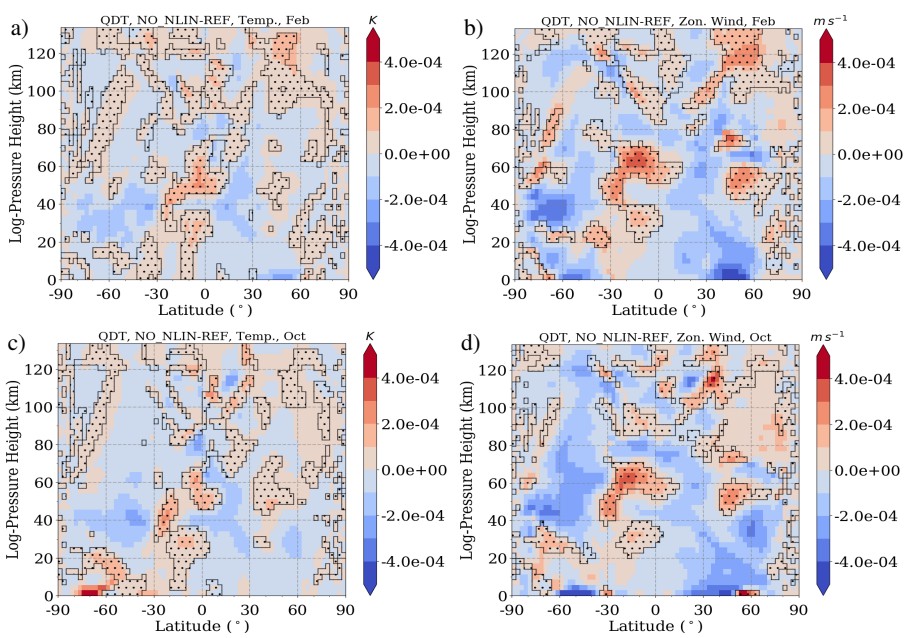

**Figure 9.** Difference of QDT amplitudes between `NO_NLIN` and `REF` simulation, scaled by $\exp\left[-z(2H)^{-1}\right]$. Red colors denote larger `NO_NLIN` simulation amplitudes and blue colors denote larger `REF` simulation amplitudes. Areas of destructive interference ($120° \leq \Delta\Phi \leq 240°$) between `NLIN` and `SOL` phases are hatched. (a, c) temperature, (b,d) zonal wind. (a, b) February conditions. (c, d) October conditions.

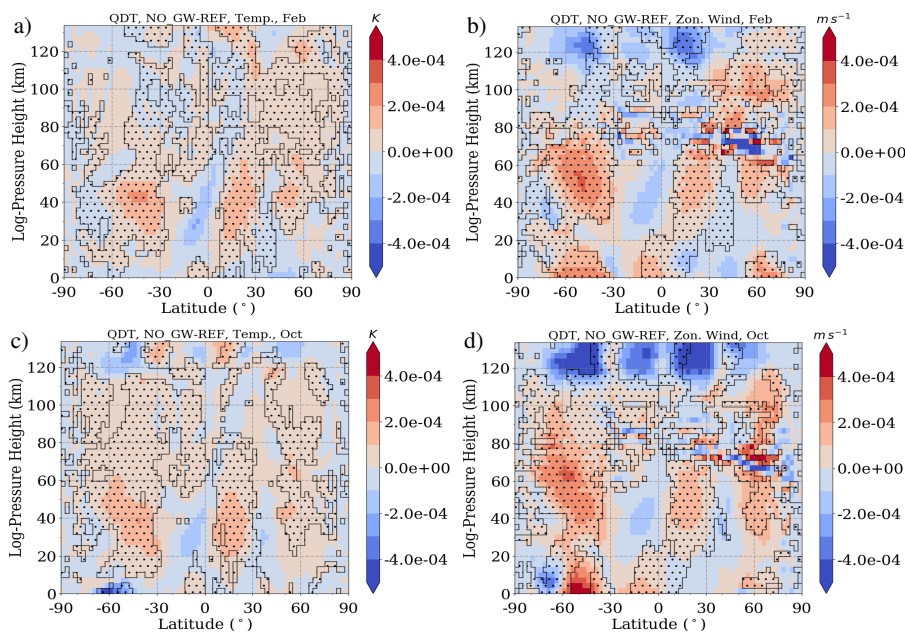

**Figure 10.** Difference of QDT amplitudes between NO_GW and REF simulation, scaled by $\exp\left[-z(2H)^{-1}\right]$. Red colors denote larger NO_GW simulation amplitudes and blue colors denote larger REF simulation amplitudes. Areas of destructive interference ($120° \leq \Delta\Phi \leq 240°$) between GW and SOL phases are hatched. (a, c) temperature, (b,d) zonal wind. (a, b) February conditions. (c, d) October conditions.