# Peer review of "Forcing mechanisms of the migrating quarterdiurnal tide"

_Annales Geophysicae, 2019_

## Referee Comment (RC1) · Anonymous Referee #1 · 29 Dec 2019

General comments:

Forcing mechanisms of the quarterdiurnal tide (QDT) have been tested by a nonlinear mechanistic global circulation model. For this, the model has been run in different configurations to analyze the importance of different forcing mechanisms (absorption of solar radiation by ozone and water vapor, nonlinear tidal interactions, and gravity wave-tide interactions). There are a few modeling studies that explore the forcing mechanisms of QDT and new insights are presented by the authors which add to our understanding of this tidal mode. The scientific contribution is appropriate for this journal, but there are some issues that need to be addressed and the language should be revised.

Specific comments:

Description of the model need to improve, mainly about gravity wave routine.

[Figure]

Page 3, line 23 - Information about the tidal forcing needs to be supplemented. Was the model run with DT, SDT, TDT and QDT modes on?

Page 5, line 2 - from Fig 1 it possible to see that largest amplitudes in the NH are found in February and October for meridional wind component whilst for zonal wind the largest amplitude appear in October for temperature in February, October and November. For the southern hemisphere, the largest amplitudes appear between April and October, being worthy of discussion.

Page 5, lines 11-12 - Please change this sentence - "The largest QDT amplitudes in the southern midlatitudes derived from the satellite data do not show agreement with the MUAM results in Fig. 1 (a)." - In fact, the model does not reproduce the observed amplitudes.

Page 7 (lines 33 and 34) and page 8 (lines 1-6): it would be interesting to separate the description NO_LIN from the description NO_GW.

Page 8, lines 19-21 and 34 - The sentences about Figs 9 and 10 are confused. Please rewrite more clearly. Are the authors dealing with the NLIN or NO_LIN case?

In Fig 1 QDT amplitudes in HS are higher than in HN (mainly in zonal wind). Could the authors discuss these differences considering the different forcing mechanisms ?

Minor/Technical comments:

Page 5, line 22 - change "bySmith et al." for "by Smith et al."

Page 5, lines 32-33 - Please provide compound term on first appearance of acronyms.

Page 5, line 22- change "(Fig. 7 g, h the amplitudes" for "(Fig. 7 g, h) the amplitudes"

In some sentences the word "model" is used insted "modeling". Please, check.

---

## Referee Comment (RC2) · Anonymous Referee #2 · 27 Jan 2020

Dear Editor,

The paper titled "Forcing mechanisms of the quarterdiurnal tide" describes a mechanistic model simulations of various forcing terms of the quarterdiurnal tide. Quarterdiurnal tide is not well studied in the past because of its small amplitudes. Consequently, we know little about its sources. Mechanistic model simulation can lead to a better understanding of these sources. The paper should be considered for publication, however, there are some issues need to addressed.

Major issues

1. The model underestimates the QDT. It will be a great help to understand the cause if we know how much the model underestimates the diurnal and semidiurnal tides. It is true that other mechanistic models also underestimate the tides. Nevertheless, there should be some discussion on this. The top height is 160 km, while the vertical

wavelength is about 100 km, is the top height sufficiently high enough for the tide?

2. When describing solar tide, the reference cited is Yigit and Medvedev (2015). While the paper may be important, it gives impression that we only found out the solar tide after 2015. Some earlier papers should be included.

Minor issues

1. I would consider add the word 'migrating' on the title.

2. P1 L8 '. . . certain seasons, latitudes, and altitudes . . .' Should be more specific.

3. P5 L16 'In addition the amplitudes of other tides (DT, SDT, TDT) are also too small compared to observations (Lilienthal, 2018)' The reference only shows the TD. There is no information on DT and SDT.

4. Figure 3 has different color scales for different terms making comparison much more difficult. Should use the same scale.

5. Figure 4 has the same issue with color scale.

6. Figures 3 4 have grey color contours, which are not easy to see. Should consider using different color.

---

## Author Comment (AC1) · 27 Jan 2020

We thank the referee for the comments. We will add additional descriptions and clarifications to the points raised. We repeat the concerns here, and add our response in italics.

Description of the model need to improve, mainly about gravity wave routine.

*The model has been extensively described, e.g., in Lilienthal et al. (2018), Samtleben et al. (2019), Fröhlich et al. (2003a, 2007), Jacobi et al. (2006). We will add more details and the respective references.*

[Figure]

Page 3, line 23 - Information about the tidal forcing needs to be supplemented. Was the model run with DT, SDT, TDT and QDT modes on?

*In the model, all tidal waves are forced through the diurnal absorption of solar radiation. We will add the following to clarify this: The diurnal cycle of solar radiation absorption leads to self-consistent forcing of tidal harmonics such as DT, SDT, TDT, and QDT.*

Page 5, line 2 - from Fig 1 it possible to see that largest amplitudes in the NH are found in February and October for meridional wind component whilst for zonal wind the largest amplitude appear in October for temperature in February, October and November. For the southern hemisphere, the largest amplitudes appear between April and October, being worthy of discussion.

*We will specify this as following: For an overview of the seasonal cycle of the QDT, Fig. 1 shows the QDT temperature and wind amplitudes at about 101km height. The amplitudes in the Northern Hemisphere show larger amplitudes in autumn and winter between $20°N - 40°N$ and $50°N - 70°N$, respectively. Maxima of QDT amplitudes are seen in the Northern Hemisphere in February and October for the meridional wind, for the zonal wind in October and for temperature in February. Larger amplitudes in the Southern Hemisphere appears also during autumn and winter (April to October) between $20°S - 40°S$ and $50°S - 70°S$ with maxima in August for temperature and winds.*

Page 5, lines 11-12 - Please change this sentence - "The largest QDT amplitudes in the southern midlatitudes derived from the satellite data do not show agreement with the MUAM results in Fig. 1 (a)." - In fact, the model does not reproduce the observed amplitudes.

*We will change this sentence to: The model results do not reproduce the amplitudes observed by satellites.*

Page 7 (lines 33 and 34) and page 8 (lines 1-6): it would be interesting to separate the description NO_LIN from the description NO_GW.

*We will change this to: In addition, a NO_NLIN run was performed in which only quarterdiurnal nonlinear interactions have been removed. The amplitudes of the NO_NLIN simulation are partly even larger than the ones in the REF simulation. This fact is also seen for the SOL simulations compared to the REF run. Larger amplitudes are also partly visible for the NO_GW simulation, with only removed interactions between tides and gravity waves in the model tendency terms.*

Page 8, lines 19-21 and 34 - The sentences about Figs 9 and 10 are confused. Please rewrite more clearly. Are the authors dealing with the NLIN or NO_LIN case?

*We will add some sentences t o this description to make it clearer: In Fig. 9, we present QDT amplitude differences between the NO_NLIN and REF simulation, which are scaled by the growth rate of the tides with altitude to highlight the actual source region of the waves. Here, the red (blue) areas denote larger amplitudes in NO_NLIN (REF) simulations. This means that in red areas the run with one removed forcing has larger amplitudes than the REF run. The removed nonlinear forcing (which is visible in the NLIN simulation) must interact with other QDT from other forcings (like solar or gravity wave forcing).*

*The NLIN run (only nonlinear forcing) case should be show small QDT amplitudes because of the weak nonlinear forcing. The NO_NLIN (without nonlinear forcing)*

*simulation should show larger amplitudes than the NLIN run, but smaller than the REF run, because one forcing (nonlinear) is missing.*

In Fig 1 QDT amplitudes in HS are higher than in HN (mainly in zonal wind). Could the authors discuss these differences considering the different forcing mechanisms ?

*This is difficult to say. The wave propagation depends on the background circulation, which of course differs between northern and southern hemisphere (temperature, and especially wind jet strength and distribution). It may be possible that the propagation conditions are better in southern hemisphere winter than in northern hemisphere winter, which leads to larger QDT amplitudes. The solar forcing depends on the ozone concentration in the respective hemisphere and these changes are small. The differences from nonlinear und gravity wave forcing between both hemispheres are also small, so that the differences in forcing cannot have the large impact that is seen in the amplitudes.*

Minor/Technical comments: Page 5, line 22 - change "bySmith et al." for "by Smith et al."

*Will be changed.*

Page 5, lines 32-33 - Please provide compound term on first appearance of acronyms.

*We will add this.*

Page 5, line 22- change "(Fig. 7 g, h the amplitudes" for "(Fig. 7 g, h) the amplitudes" In some sentences the word "model" is used insted "modeling". Please, check.

*That's right, we will correct this.*

*We will correct also another mistake on page 3, line 19-21:*

*Old: In the ensemble runs, the **ozone mixing ratio** is chosen according to the Mauna Loa Observatory data for 2005 (e.g., 380 ppm for February 2005, [...]), because we do not intend to perform an **ozone** dependent trend analysis.*

*New: In the ensemble runs, the **$CO_2$ concentration** is chosen according to the Mauna Loa Observatory data for 2005 **as global constant** (e.g., 380 ppm for February 2005, [...]), because we do not intend to perform an **ozone and $CO_2$** dependent trend analysis.*

---

## Author Comment (AC2) · 13 Feb 2020

We thank the referee for the comments. We will add additional descriptions and clarifications to the points raised. We repeat the concerns here, and add our response in italics.

The model underestimates the QDT. It will be a great help to understand the cause if we know how much the model underestimates the diurnal and semidiurnal tides. It is true that other mechanistic models also underestimate the tides. Nevertheless, there should be some discussion on this. The top height is 160 km, while the vertical wavelength is about 100 km, is the top height sufficiently high enough for the tide?

*We will add graphs from the current model version for terdiurnal, diurnal and semid-*

[Figure]

*iurnal tides to the supplement and discuss this problem in more detail. The forcing of the QDT takes place up to a height of about 70 km for the important solar forcing. The wavelength is of the order of 100 km, but becomes substantially smaller at higher altitude (where nonlinear and gravity interaction force QDT), therefore the model height is most probably sufficient for the simulation of the QDT.*

When describing solar tide, the reference cited is Yigit and Medvedev (2015). While the paper may be important, it gives impression that we only found out the solar tide after 2015. Some earlier papers should be included.

*We will add some earlier papers.*

I would consider add the word 'migrating' on the title.

*We will add the word "migrating".*

P1 L8 '...certain seasons, latitudes, and altitudes...' Should be more specific.

*We will go into more detail here.*

P5 L16 'In addition the amplitudes of other tides (DT, SDT, TDT) are also too small compared to observations (Lilienthal, 2018)' The reference only shows the TD. There is no information on DT and SDT.

*We will specify the discussion more and supplement the graphics of the DT and SDT in this paper.*

Figure 3 has different color scales for different terms making comparison much more difficult. Should use the same scale.
Figure 4 has the same issue with color scale.

*The scales in the figure will be adjusted again and standardized.*

Figures 3, 4 have grey color contours, which are not easy to see. Should consider using different color.

*It is difficult to find a contour color that is easy to recognize in all cases. We will try to improve this.*

---

## Author Response (AR1)

We thank the referees for the comments. We will add additional descriptions and clarifications to the points raised. We repeat the concerns here, and add our response in italics.

Answers to Reviewer 1:

Description of the model need to improve, mainly about gravity wave routine.

*The model has been extensively described, e.g., in Lilienthal et al. (2017, 2018), Samtleben et al. (2019), Fröhlich et al. (2003a, 2007), Jacobi et al. (2006). We have added more references to the gravity wave routine description: Yigit et al. (2008, 2009).*

Page 3, line 23 - Information about the tidal forcing needs to be supplemented. Was the model run with DT, SDT, TDT and QDT modes on?

*In the model, all tidal waves are forced through the diurnal absorption of solar radiation. We added the following to clarify this: The diurnal cycle of solar radiation absorption leads to self-consistent forcing of tidal harmonics such as DT, SDT, TDT, and QDT.*

Page 5, line 2 - from Fig 1 it possible to see that largest amplitudes in the NH are found in February and October for meridional wind component whilst for zonal wind the largest amplitude appear in October for temperature in February, October and November. For the southern hemisphere, the largest amplitudes appear between April and October, being worthy of discussion.

*We have specified this as following: For an overview of the seasonal cycle of the QDT, Fig. 1 shows the QDT temperature and wind amplitudes at about 101 km height. In the northern hemisphere, amplitudes increase in autumn and winter in the latitude ranges 20°N-40°N and 50°N-70°N, respectively. Maximum wind amplitudes in the northern hemisphere are seen in February and October; for the meridional wind largest amplitudes are found in the 20°N-40°N range, while zonal wind and temperature QDT amplitudes during these months are seen at 50°N-70°N.*
*In the southern hemisphere maximum amplitudes appear also during autumn and winter (April to October) between 20°S-40°S and 50°S-70°S. The higher latitude maximum is more strongly expressed than in the northern hemisphere.*

Page 5, lines 11-12 - Please change this sentence - "The largest QDT amplitudes in the southern midlatitudes derived from the satellite data do not show agreement with the MUAM results in Fig. 1 (a)." - In fact, the model does not reproduce the observed amplitudes.

*We have changed this sentence to: Our simulated maximum in October, on the other hand, does not appear in the SABER/TIMED data. Also, the extrema at about 10°N in June, September and October as reported by Liu et al. (2015) do not match with the MUAM results, because the amplitudes in the model are much smaller than the ampli-*

*tudes observed by satellites.*

Page 7 (lines 33 and 34) and page 8 (lines 1-6): it would be interesting to separate the description NO_LIN from the description NO_GW.

*We have changed this to: In addition, a NO_ NLIN run was performed in which only quarterdiurnal nonlinear interactions have been removed. The amplitudes of the NO_ NLIN simulation are partly even larger than the ones in the REF simulation. This fact is also seen for the SOL simulations compared to the REF run. Larger amplitudes are also partly visible for the NO_ GW simulation, with only interactions between tides and gravity waves removed in the model tendency terms. The amplitudes (Fig. S1 and S3) and phases (Fig. S2 and S4) of these simulations are shown in the supplement, because amplitude and phase differences compared to the REF simulation are rather small.*

Page 8, lines 19-21 and 34 - The sentences about Figs 9 and 10 are confused. Please rewrite more clearly. Are the authors dealing with the NLIN or NO_LIN case?

*We have specified this: In Fig. 9, we present QDT amplitude differences between the NO_ NLIN and REF simulation, which are scaled with density to highlight the actual source region of the waves. Here, the red (blue) areas denote larger amplitudes in NO_ NLIN (REF) simulations. This means that in red areas the run with one removed forcing has larger amplitudes than the REF run. We conclude that the removed nonlinear forcing must have destructively interfered with other QDT from other forcings (like solar or gravity wave forcing). The NLIN run (only nonlinear forcing) case is expected to show small QDT amplitudes because of the weak nonlinear forcing. Without destructive interference, the NO_NLIN (without nonlinear forcing) simulation should show larger amplitudes than the NLIN run, but smaller ones than the REF run, because one forcing (nonlinear) is missing.*

In Fig 1 QDT amplitudes in HS are higher than in HN (mainly in zonal wind). Could the authors discuss these differences considering the different forcing mechanisms?

*This is difficult to say. The wave propagation depends on the background circulation, which of course differs between northern and southern hemisphere (temperature, and especially wind jet strength and distribution). It may be possible that the propagation conditions are better in southern hemisphere winter than in northern hemisphere winter, which leads to larger QDT amplitudes. The solar forcing depends on the ozone concentration in the respective hemisphere and these changes are small. The differences from nonlinear und gravity wave forcing between both hemispheres are also small, so that the differences in forcing cannot have the large impact that is seen in the amplitudes.*

Minor/Technical comments: Page 5, line 22 - change "bySmith et al." for "by Smith et al."

*Has been changed.*

Page 5, lines 32-33 - Please provide compound term on first appearance of acronyms.

*We have added this.*

Page 5, line 22- change "(Fig. 7 g, h the amplitudes" for "(Fig. 7 g, h) the amplitudes"
In some sentences the word "model" is used insted "modeling". Please, check.

*This is corrected.*

*We have corrected also another mistake on page 3, line 19-21: Old: In the ensemble runs, the ozone mixing ratio is chosen according to the Mauna Loa Observatory data for 2005 (e.g., 380 ppm for February 2005, [...]), because we do not intend to perform an ozone dependent trend analysis. New: We do not intend to perform an ozone and $CO_2$ dependent trend analysis, so we leave both constant in all simulations. Ozone is implemented as monthly mean zonal mean field for the year 2005 up to 50 km altitude taken from MERRA-2 (Modern-Era Retrospective Analysis for Research and Application, version 2) reanalysis data (MERRA, 2019 and Gelaro et al., 2017) for each of the ensemble members. Above 50 km, the ozone mixing ratio is assumed to decrease exponentially. In the ensemble runs, the $CO_2$ mixing ratio is chosen according to the Mauna Loa Observatory data for 2005 as global constant up to 80 km and an exponential decrease above (e.g., 380 ppm for February 2005; NOAA,2018 and Thoning et al., 1989).*

Answers to Reviewer 2:

The model underestimates the QDT. It will be a great help to understand the cause if we know how much the model underestimates the diurnal and semidiurnal tides. It is true that other mechanistic models also underestimate the tides. Nevertheless, there should be some discussion on this. The top height is 160 km, while the vertical wavelength is about 100 km, is the top height sufficiently high enough for the tide?

*We have added graphs from the current model version for diurnal and semidiurnal tides to the supplement and discuss this problem in more detail. The forcing of the QDT takes place up to a height of about 70 km for the important solar forcing. The wavelength is of the order of 100 km, but becomes substantially smaller at higher altitude (where non-linear and gravity interactions force QDT), therefore the model height is most probably sufficient for the simulation of the QDT.*

When describing solar tide, the reference cited is Yigit and Medvedev (2015). While the paper may be important, it gives impression that we only found out the solar tide after 2015. Some earlier papers should be included.

*We have added some earlier papers.*

I would consider add the word 'migrating' on the title.

*"migrating" is added to the title.*

P1 L8 '...certain seasons, latitudes, and altitudes...' Should be more specific.

*Changed this to: "...at lower and middle latitudes in the mesosphere and lower thermo-sphere."*

P5 L16 'In addition the amplitudes of other tides (DT, SDT, TDT) are also too small compared to observations (Lilienthal, 2018)' The reference only shows the TD. There is no information on DT and SDT.

*We added a brief discussion to this topic and put the graphics of the DT and SDT in the supplement. Fig. S5 in the supplement shows the DT and SDT zonal wind amplitudes for February and October from the MUAM REF simulation and the climatology from the Global Scale Wave Model (GSWM, 2020). The latitude-dependent structure and the increase of the amplitudes with altitude is correctly reproduced by the MUAM model. However, the maxima of the DT and SDT amplitudes in MUAM are at low latitudes and midlatitudes sometimes more than 50% lower than those of the GSWM. Also comparison with radar measurements (e.g. Manson et al., 1989, Pokhotelov et al., 2018) shows that the amplitudes of the DT and SDT are underestimated in MUAM.*

Figure 3 has different color scales for different terms making comparison much more difficult. Should use the same scale.
Figure 4 has the same issue with color scale.

*The scales in the figure are changed and standardized.*

Figures 3, 4 have grey color contours, which are not easy to see. Should consider using different color.

*We used a different color and hope it improved the visibility.*

All relevant changes in the manuscript are already listed above.

[revised manuscript text omitted]